# Cu_14_H_12_(P*^t^*Bu_3_)_6_Cl_2_—The Expanse of Stryker’s Reagent

**DOI:** 10.3390/molecules30244779

**Published:** 2025-12-15

**Authors:** Markus Strienz, Roman Kimmich, Alexander Conzelmann, Andreas Schnepf

**Affiliations:** Institut für Anorganische Chemie, Universität Tübingen, Auf der Morgenstelle 18, 72076 Tübingen, Germany

**Keywords:** copper hydride, hydrogenation, Stryker’s reagent, mass spectrometry, neuronal network

## Abstract

Although a large number of copper hydride complexes and clusters have been reported, phosphine-stabilized copper hydrides remain comparatively rare. This is particularly noteworthy given the continuing interest in Stryker’s reagent [HCu(PPh_3_)]_6_ due to its use as a hydrogenation reagent. In this work, we report on the synthesis and full characterization of a novel copper hydride cluster, [Cu_14_H_12_(P*^t^*Bu_3_)_6_Cl_2_]. The structure of this copper hydride was determined via SC-XRD. In addition, the reactivity of the hydrides and their position were investigated via a convolutional neural network, quantum chemical calculations, and NMR, and they are compared to the well-known, smaller Stryker’s reagent.

## 1. Introduction

Selective hydrogenation, especially of molecules bearing additional functional groups, has long been a sought-after goal [1,2,3,4]. Therefore, it is not surprising that with Stryker’s reagent, a hydrogenation reagent has been chosen to be the chemical compound of the year 1991 [5]. Stryker’s reagent, [HCu(PPh_3_)]_6_ (**1**), selectively hydrogenates α,β-unsaturated aldehydes, ketones, and esters, and reductive silylation is reported as well. These reactions are possible in the presence of other C–C double bonds, leading to an excellent selectivity for Stryker’s reagent. Stryker’s reagent has also been successfully employed in the synthesis of metal clusters [6,7].

Its six copper atoms are arranged in the form of a distorted octahedron consisting of six smaller and two larger triangular faces (see Figure 1). The positions of the hydrides have been discussed since its first synthesis in 1971 and later determined to be over the six smaller triangles of the octahedron, slightly distorted to one edge [8,9,10,11,12]. Each copper atom is coordinated by PPh_3_. Analogues of Stryker’s reagent have also been synthesized, with P(tol)_3_ and P(NMe_2_)_3_ as ligands, and both compounds show a similar distorted octahedral core [9,10]. While there is a large number of copper complexes and clusters stabilized by chalcogenides and alkynes [13,14,15], there are only few examples that exhibit exclusively phosphines and hydrides and can therefore be seen as an extension of Stryker’s reagent [9,16]. Here, we present the novel copper hydride compound [Cu_14_H_12_(P*^t^*Bu_3_)_6_Cl_2_] (**2**), with a distorted face-capped-cubic arrangement of the copper atoms, and investigate it with respect to its structure, the position of the hydrides, and its potential usage as a hydrogenation reagent.

## 2. Results

The title compound [Cu_14_H_12_(P*^t^*Bu_3_)_6_Cl_2_] (**2**) is obtained from the reaction of (*^t^*Bu_3_P)CuCl with L-Selectride (LiHB*^s^*Bu_3_) in tetrahydrofuran (THF). After extraction with pentane or cyclopentane, **2** crystallizes at –30 °C in the form of orange crystals in the triclinic space group *P*$\bar{1}$ (see Figure 2). Within the crystals, the solvent used for the extraction co-crystallizes. **2** is stable in air for a few weeks and can be stored at room temperature under inert conditions.

The composition of **2** is determined via SC-XRD (single-crystal X-ray diffraction) and ESI-MS (electrospray ionization mass spectrometry) measurements. Therefore, the molecular structure of **2** can be described as follows: Eight of the twelve copper atoms form a distorted cube in the center of the cluster, featuring six shorter (250.1 ± 0.3 pm) and six longer (264.6 ± 0.5 pm) edges (see Figure 3a). The six elongated Cu–Cu distances involve Cu4 or Cu4′ and are positioned along a body diagonal to the cube, indicating a distortion along this axis (see Figure 3a). Cu4 and Cu4′ each bind to a chloride with a Cu–Cl bond length of 229.7 pm. Each of the six faces of the Cu_8_ cube is capped by a Cu(P*^t^*Bu_3_) unit, forming six pyramidal structures that share the edges of the cube’s bottom face (see Figure 3b). This structural arrangement is also known as a tetrakis hexahedron. The Cu–Cu distances between the Cu_8_ cube and the bridging Cu(P*^t^*Bu_3_) units range from 247.6 pm to 271.6 pm. The longer distances, like those within the Cu_8_ core, involve Cu4 and Cu4′ (270.6 ± 0.9 pm). The six Cu(P*^t^*Bu_3_) units form an octahedron reminiscent of Stryker’s reagent and can be viewed as a spatial extension of this [HCu(PPh_3_)]_6_ complex described in 1971 [8]. In addition to the structure, the Cu–Cu distances are also in the same range for both compounds ([HCu(PPh_3_)]_6_: 249.4–267.4 pm). However, the smaller copper hydride compound lacks CuCl units. This may be one reason for the relatively low yield of **2**, since the correct stoichiometry between CuH and CuCl is necessary.

A similar arrangement of metal atoms is observed within the metalloid cluster Ga_14_(Si(SiMe_3_)_3_)_6_ [17]. Comparable structural motifs are also found in the two Al_14_SiR_6_ clusters (R = Cp* [18], N(Dipp)SiMe_3_ [19]) and in the two Sn_15_R_6_ clusters (R = N(Dipp)SiMe_3_, N(Dipp)SiMe_2_Ph) [20]. However, in contrast to **2** and the Ga_14_ cluster, the central atom position of the aluminum and tin clusters is occupied, leading to a total number of 15 metal atoms.

Multinuclear copper clusters, particularly those synthesized from hydride sources, often incorporate hydrides within their structures. However, accurately determining the number and positions of the hydrides in such copper clusters remains a challenging task, owing to the limitations of current detection methods. While the positions of hydrides in copper chalcogenide compounds are often located inside the metal cage [21,22], the hydrogen atoms in Stryker’s reagent are located on the surface of the Cu_6_ octahedron [9,10]. In **1**, six of the eight faces of the octahedron are capped by hydrides in a μ_3_-fashion. The capping of only six faces leads to the observed distortion of the Cu_6_ core, with the two uncapped faces positioned opposite each other along a diagonal. This has been confirmed by SC-neutron diffraction [10], powder neutron diffraction [11], and micro-SC-electron diffraction [12]. Since neutron diffraction requires a cost-intensive setup and large single crystals, neural networks become a powerful tool in determining the positions of hydrogen atoms in copper compounds [23,24].

To determine the number of the hydrides in compound **2**, ESI-MS was performed after dissolving the compound in acetonitrile (MeCN) (Figure 4).

The ESI-MS spectrum displays a signal at *m/z* = 2141.18, which corresponds to the monoisotopic mass of [Cu_14_H_12_(P*^t^*Bu_3_)_6_Cl]^+^ (see Figure 4, top). The experimental isotopic pattern is in excellent agreement with the simulated spectrum (*m/z* = 2141.19) (see Figure 4, bottom). The cationic species results from the loss of one chloride ligand during the ionization process. Additionally, the twofold positively charged species, [Cu_14_H_12_(P*^t^*Bu_3_)_6_]^2+^, is observed at *m/z* = 1053.1 (see Appendix A).

The twelve hydrides, together with the two chlorides, result in an oxidation state of +I for all copper atoms. Therefore, despite the structural similarity, **2** differs from the metalloid clusters Ga_14_, Al_14_Si, and Sn_15_, in which some of the metal atoms exhibit the oxidation state ±0 [17,18,19,20].

The number of hydrides obtained via ESI-MS is in accordance with the results obtained via proton NMR. In the ^1^H-NMR spectrum, two signals with a combined integral of 12 (relative to the 162 protons of the phosphine ligands) are observed (see Figure 5). The signals of the 12 hydrides can be found at 3.10 ppm and 3.92 ppm in a ratio of 1:1, indicating two chemically inequivalent hydride positions. The chemical shift for the hydride atoms is in a comparable range to those found in Stryker’s reagent [HCu(P(tol)_3_]_6_: 3.5 ppm [25]).

To determine the positions of the twelve hydrides, we used a convolutional neural network trained on a dataset of the structures of 25 copper hydride compounds acquired by neutron diffraction experiments. A similar approach was used by Wang et al. [24].

For the dataset, we project each copper hydride compound onto a grid of size [42.5, 42.5, 42.5] Å. This grid is subdivided into 512 voxels (a 3D equivalent of a pixel) per dimension. We then combine the voxels together to non-overlapping patches of size [32, 32, 32] and label them based on whether they contain a hydride or not. To predict the label, we train a 3D convolutional neural network, for which the labeled patches serve as training data. As our patches are highly imbalanced (i.e., there are much more patches which do not contain a hydride than ones that do), we use the focal loss for training [26]. To evaluate our network, we perform the same preprocessing as for our training dataset with our target copper compound. We then let the network predict for each patch whether it contains a hydrogen atom or not, which it does for 12 patches, in accordance with our expectations.

The predicted hydrogen atoms tend to cap Cu_3_ triangles in a μ_3_ arrangement. This is similar to the hydride position in **1**, but in contrast to other copper hydride compounds like [Cu_11_H_2_(S_2_P(O*^i^*Pr)_2_)_6_(C≡CR)_3_] (R = Ph, C_6_H_4_F, and C_6_H_4_OMe), which show also hydrides in the cluster core [22]. One reason could be the absence of thiolates, which span over faces. In contrast, the phosphines and chlorides in **1** and **2** are all terminally coordinated, allowing for the bridging of the hydrides over faces. Using the knowledge of the hydride positions from the CNN, we placed the 12 hydrides over 12 triangular faces of **2**. Geometrical optimization via DFT gives a minimum structure, which preserves the overall structure, similar to the experimental one. The cubic core has six shorter (251.9 ± 0.5 pm) and six longer (267.2 ± 0.7 pm) Cu–Cu distances, almost identical to the experimentally obtained structure (250.1 ± 0.3 pm, 264.6 ± 0.5 pm). The distances between the copper atoms of the cubic core and the capping copper–phosphine unit range from 244.7 pm to 269.5 pm (exp: 247.6 pm to 271.6 pm). The difference between the Cu–Cu bonds results from their different neighboring atoms. Additional to the bound chlorine atoms, the hydrides cap only two of the four triangles. The twelve hydrides bind to the copper atoms in a manner that half of the Cu_3_ triangles of each of the six Cu_5_ pyramids are capped (Figure 6). One of the two hydrides of each pyramid is in proximity to the CuCl unit. Therefore, the hydrides are chemically inequivalent in a 1:1 ratio, as confirmed by NMR.

### Reactivity Investigation

As already mentioned, Stryker’s reagent was awarded chemical compound of the year in 1991 due to its exceptional selective hydrogenation of α,β-unsaturated C–C double bonds [5]. Prominent examples include the hydrogenation of cyclohex-2-en-1-one and benzalacetone, in which only the C–C double bond is reduced, while the carbonyl group remains unaffected. Additionally, no reaction was observed with double bonds such as those in ethylene or acetylene, suggesting a partial involvement of the copper atoms.

To evaluate the reactivity of **2**, which can be seen as a direct structural analogue of Stryker’s reagent, we reacted compound **2** with an excess of cyclohex-2-en-1-one in benzene-d_6_ (Figure 1).

The successful hydrogenation was confirmed by the formation of cyclohexanone, as monitored by ^1^H-NMR spectroscopy (see Appendix A). **2** acts as a milder hydrogenation reagent compared to Stryker’s reagent. While the reaction of cyclohex-2-en-1-one and Stryker’s reagent takes 15 min at room temperature, we observed the full conversion after 3 days at 60 °C. A second hydrogenation reaction of compound **2** with an excess of dimethyl fumarate afforded dimethyl succinate (Figure 1). The progress of the reaction was monitored by ^1^H-NMR spectroscopy over the course of one day (see Figure 7).

The signals observed in the spectra from 1.0 to 3.3 ppm illustrate the consumption of compound **2** and dimethyl fumarate, as well as the simultaneous formation of dimethyl succinate. The signal at 1.56 ppm, corresponding to the *tert*-butyl group, decreases over time and disappears completely after 21 h. Simultaneously, three new signals appear at 1.23 ppm, 2.29 ppm, and 3.26 ppm, gradually increasing in intensity. The signals at 2.29 ppm and 3.26 ppm correspond to the hydrogenated compound dimethyl succinate, exhibiting an integral ratio of 6:4 (see Appendix A). The doublet at 1.23 ppm can be attributed to an unknown compound containing *tert*-butyl phosphine groups. The chemical shift at 1.23 ppm differs from that of free tri-*tert*-butyl phosphine (P*^t^*Bu_3_, 1.27 ppm) and the precursor (*^t^*Bu_3_P)CuCl (1.31 ppm). Based on the observed color change in the solution from bright orange to brown, as well as a hydrodynamic radius of 9.1 nm determined by dynamic light scattering (DLS), we propose the formation of phosphine-stabilized copper nanoparticles (see Appendix A).

In contrast to the two successful hydrogenation reactions described above, no reaction was observed with ethylene.

## 3. Materials and Methods

L-Selectride (1.0 M in THF) was purchased from Sigma Aldrich (St. Louis, MO, USA) and used without further purification. Tetrahydrofuran and cyclopentane were dried over Na/Benzophenone; pentane was dried over CaH_2_ and distilled before further use. All reactions were carried out under an argon atmosphere.

(*^t^*Bu_3_P)CuCl was synthesized according to literature procedure from *^t^*Bu_3_P and CuCl [27].

Synthesis of [Cu_14_(P*^t^*Bu_3_)_6_Cl_2_H_12_]: (*^t^*Bu_3_P)CuCl (301.3 mg, 1.0 mmol, 1.0 equiv.) was dissolved in THF (30 mL), and L-Selectride (0.7 mL, 1.0 M solution in THF, 0.7 equiv.) was added dropwise under vigorous stirring. The reaction mixture was stirred at room temperature for 1.5 h. Volatile components were removed in vacuo, and the resulting solid was extracted with pentane (20 mL). After concentration and storage at −30 °C, the product **2** was obtained as orange crystals (15.3 mg, 6.9 μmol, 9.6%). ^1^H-NMR (400.1 MHz, C_6_D_6_): δ [ppm] = 3.92 (s, 6 H, Cu-*H*), 3.10 (s, 6 H, Cu-*H*), 1.58 (d, 162 H, ^3^J*_PH_* = 12.0 Hz, C*H_3_*), ^31^P{^1^H}-NMR (162.0 MHz, C_6_D_6_): δ [ppm] = 62.7 (s, P*^t^*Bu_3_), ^13^C-NMR (100.6 MHz, C_6_D_6_): δ [ppm] = 32.9 (d, ^2^J*_PC_* = 6.4 Hz, *C*H_3_).

NMR-spectra were measured using a Bruker AVIIIHD-300, AVII+400, or AVIII600 spectrometer (Billerica, MA, USA). The chemical shifts are given in ppm against the external standard SiMe_4_. C_6_D_6_ was dried with 3 Å molecular sieve.

Single crystals of compound **2** were measured with a Bruker APEX II DUO diffractometer equipped with an IμS microfocus sealed tube and QUAZAR optics for monochromated Mo_Kα_ radiation (λ = 0.71073 Å) at 100 K. Absorption correction was applied using the program SADABS 2016/2 [28]. The structure was solved by direct methods and refined against *F*^2^ for all observed reflections. Programs used: SHELXT 2014/5 and SHELXL 2019/3 within the OLEX2 V1.5 program package [29,30,31,32]. The supplementary crystallographic data (CCDC 2504326) can be obtained online free of charge at https://www.ccdc.cam.ac.uk/structures/ (accessed on 21 November 2025) or from Cambridge Crystallographic Data Centre, 12 Union Road, Cambridge CB21EZ; Fax: (+44) 1223-336-033; or deposit@ccdc.cam.ac.uk.

Mass spectrometry data were obtained at a HR-ESI-TOF (maXis 4G, Bruker).

UV/Vis spectra were measured using a T60 UV/Vis-spectrometer from PG Instruments (Leicestershire, UK) in a 1 cm quartz glass cuvette. As a reference, a 1 cm glass cuvette with pure solvent was used.

DLS measurements were conducted with a Zetasizer Nano ZS spectrometer from Malvern Instruments (Worcestershire, UK).

DFT calculations were conducted with the program package TURBOMOLE (V 7.4.1) [33] using the GUI TmoleX (V 4.4.1) [34]. Geometrical optimizations were conducted, starting from the experimentally obtained molecular structure based on the BP86-D3BJ/def2-SV(P)//PBE0-D3BJ/def2-TZVP level of theory.

The voxel values of the processed data for the neural network are set as follows: First, we assign each element that is present in the copper hydride compound to one channel. Then, we iterate over each atom in the compound and use a quadratical decay based on the radius of the atom to set the intensity values of all other voxels in the corresponding channel. After labeling the patches, we removed all patches whose voxel values did not sum up to a minimum threshold value of 2.0. For the neural network, we use 3D convolutions with channel sizes [7, 16, 32, 64, 128], followed by a single linear binary classification layer. Each convolutional layer is followed by a max pooling layer, and we use dropout on the linear layer at a rate of 0.05. We train for 20 epochs using AdamW at the default settings. All machine learning code was written using the PyTorch (V2.7.1) framework, and our code is available under the following github: https://github.com/Conzel/cu14-hydride-prediction (accessed on 21 November 2025).

## 4. Conclusions

To summarize the results, the reaction of CuCl with 0.7 equivalents of L-Selectride yields in the presence of P*^t^*Bu_3_ the cluster compound [Cu_14_H_12_(P*^t^*Bu_3_)_6_Cl_2_] (**2**). **2** exhibits structural motifs like those of Stryker’s reagent **1** in form of six octahedrally arranged Cu(P*^t^*Bu_3_) units bound to the Cu_8_ cube via μ_3_-capping hydrides. The structural similarity results in a comparable chemical behavior, allowing compound **2** to be used as a hydrogenating agent for electron–poor carbon–carbon double bonds. This makes **2** an expansion of Stryker’s reagent and potentially the first member of a series of analogues containing additional CuCl units, stabilized by sterically demanding phosphine ligands.

## Data Availability

The original contributions presented in this study are included in the article/Appendix A. Further inquiries can be directed to the corresponding author.

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
