# Peer review of "Cu_14_H_12_(P*^t^*Bu_3_)_6_Cl_2_—The Expanse of Stryker’s Reagent"

_molecules, 2025, doi:10.3390/molecules30244779_

Round 1
Reviewer 1 Report
Comments and Suggestions for Authors
The authors report the synthesis and characterization of a new phosphine-stabilized copper hydride cluster, Cu14H12(PtBu3)6Cl2, structurally related to Stryker’s reagent, and use a CNN+DFT approach to assign the hydride positions, followed by some hydrogenation tests. The structure is clearly established, the idea of using a trained model for hydride placement is timely, and the manuscript is generally well written and suitable for Molecules. I recommend minor revision.
1 Since the CNN-assisted placement of 12 μ3-hydrides is central to the paper, I would like to see a bit more comment on how reliable the protocol is. Even 2-3 sentences in the main text or SI would help readers judge how robust the proposed model is.
2 The presence of two hydride signals in a 1:1 ratio is interesting in light of the final symmetric hydride model, and it would be helpful if the authors could briefly explain what structural features generate these two environments.
3 Given that 2 is isolated in modest yield, it would be useful to briefly comment on whether the preparation is reproducible on slightly larger scale and how stable the cluster is in the solid state and in solution. One or two sentences are enough, but they will be valuable for readers who may want to use 2 as a reagent.
4 The “expanse of Stryker’s reagent” concept is attractive; I would encourage the authors to add a short, more quantitative comparison to underline what is genuinely new about this architecture.
Author Response
The authors report the synthesis and characterization of a new phosphine-stabilized copper hydride cluster, Cu14H12(PtBu3)6Cl2, structurally related to Stryker’s reagent, and use a CNN+DFT approach to assign the hydride positions, followed by some hydrogenation tests. The structure is clearly established, the idea of using a trained model for hydride placement is timely, and the manuscript is generally well written and suitable for Molecules. I recommend minor revision.
Answer: Thank you very much for your overall positive assessment of our work.
1 Since the CNN-assisted placement of 12 μ3-hydrides is central to the paper, I would like to see a bit more comment on how reliable the protocol is. Even 2-3 sentences in the main text or SI would help readers judge how robust the proposed model is.
Answer: Thank you for pointing this out. During the design of our neural network, we noticed that the approach is particularly sensitive to the data distribution, i.e. how many cells contain a hydride and how many do not. Since the clusters contain only few hydrides compared to the massive number of cells that can fit into our grid, if no careful counter-measures are taken, the network is incentivized to always predict ‘no hydride’. We offset this via three crucial measures, namely by 1) using the focal loss, which is designed for imbalanced datasets, 2) filtering out all patches which are far away from other atoms (measured by the summed voxel intensity of the patch against a threshold value) and 3) carefully choosing a grid size that is neither too small (skewing the ratio between no-hydride to hydride-containing patches) or too large (losing prediction accuracy). Once the data distribution is correctly set, we find our approach to work well with wide varieties of hyperparameters (e.g. learning rate, dropout rate) and neural network architectures (number of layers and filter sizes), requiring little additional tuning.
2 The presence of two hydride signals in a 1:1 ratio is interesting in light of the final symmetric hydride model, and it would be helpful if the authors could briefly explain what structural features generate these two environments.
Answer: Thank you for this hint and we added a sentence in the manuscript explaining that the 1:1 ratio of the hydrides arises from the proximity of one hydride to the CuCl unit.
3 Given that 2 is isolated in modest yield, it would be useful to briefly comment on whether the preparation is reproducible on slightly larger scale and how stable the cluster is in the solid state and in solution. One or two sentences are enough, but they will be valuable for readers who may want to use 2 as a reagent.
Answer: Thanks for this hint. For the formation of 2, a 2:12 ratio of CuH to CuCl is necessary. While incomplete reaction of CuCl to CuH is prevented due to the 0.7 equivalents of L-Selectride, there are still variations in local concentrations. Decreasing the amount of L-Selectride further reduces the yield of 2, since there are not enough hydrides available. However, we have not tested an upscaling.
Compound 2 exhibits high stability and remains stable in the crystalline state under air for several weeks. Similarly, 2 is stable in solution. We have added a sentence about this to the manuscript.
4 The “expanse of Stryker’s reagent” concept is attractive; I would encourage the authors to add a short, more quantitative comparison to underline what is genuinely new about this architecture.
Answer: We added a brief comparison of the Cu–Cu distances in both compounds.
Reviewer 2 Report
Comments and Suggestions for Authors
The MS by Schnepf et al. describes a new copper hydride formulated as [Cu14H12(PtBu3)6Cl2]. The complex was characterized by SCXRD, NMR and mass-spec. Its catalytic activity in hydrogenation reactions was tested by hydrogenation of cyclohex-2-en-1-one and dimethyl fumarate. The search of H-atoms positions in the crystal structure was supported by AI which is quite interesting. At the moment, such approaches are unique and should be highlighted. The reported data are in the scope of Molecules and can be published after taking these minor issues into account:
1) According to the coordination chemistry rules, all coordination compounds should be written in square brackets
2) In the SI authors indicate the presence of pentane as solvate molecule, while there is nothing about this in the main text. So, the formula of 2 should be corrected.
3) Full mass-spec data should be given in SI according to the standards.
Comments on the Quality of English LanguageThe English could be improved to more clearly express the research.
Author Response
The MS by Schnepf et al. describes a new copper hydride formulated as [Cu14H12(PtBu3)6Cl2]. The complex was characterized by SCXRD, NMR and mass-spec. Its catalytic activity in hydrogenation reactions was tested by hydrogenation of cyclohex-2-en-1-one and dimethyl fumarate. The search of H-atoms positions in the crystal structure was supported by AI which is quite interesting. At the moment, such approaches are unique and should be highlighted. The reported data are in the scope of Molecules and can be published after taking these minor issues into account:
Answer: Thanks for this overall positive rating of our work.
1) According to the coordination chemistry rules, all coordination compounds should be written in square brackets
Answer: We thank the reviewer for this comment and have made the corresponding changes.
2) In the SI authors indicate the presence of pentane as solvate molecule, while there is nothing about this in the main text. So, the formula of 2 should be corrected.
Answer: We have added the co-crystallizing solvent to the main text.
3) Full mass-spec data should be given in SI according to the standards.
Answer: We added all necessary mass-spec data in the SI. Thank you for the comment, we missed an important point there.
Comments on the Quality of English Language
The English could be improved to more clearly express the research.
Answer: We have reworked the manuscript, revised sentences where we felt improvements were needed, and corrected some spelling errors.
Reviewer 3 Report
Comments and Suggestions for Authors
The manuscript reports the synthesis and characterization of Cu₁₄H₁₂(PᵗBu₃)₆Cl₂, a novel copper hydride cluster that represents a structural extension of Stryker’s reagent [HCu(PPh₃)]₆, a well-established selective hydrogenation agent. Using SC-XRD, ESI-MS, NMR, and a convolutional neural network combined with DFT calculations, the authors elucidate its architecture: a distorted Cu₈ cube capped by six Cu(PᵗBu₃) units and twelve μ₃-hydrides. The compound demonstrates selective hydrogenation of electron-deficient C=C bonds (e.g., cyclohex-2-en-1-one, dimethyl fumarate) while remaining inactive toward ethylene, closely resembling the reactivity profile of Stryker’s reagent. Overall, this work provides a significant contribution to the development of new structural motifs in multinuclear copper hydride chemistry and highlights its potential as an expanded analogue of Stryker’s reagent. The manuscript is suitable for publication in Molecules after addressing the following minor points:
- Page 6: The authors evaluate the reactivity of compound 2; however, the reaction scheme should be included in the main text. Additionally, a direct comparison of its reactivity with reported Stryker’s reagent analogues is necessary.
- Page 8: The reported yield of compound 2 is quite low. What is the major product of the reaction? Please provide data on the crude mixture in the Supporting Information.
- What is the outcome when (tBu₃P)CuCl reacts with an excess of L-Selectride?
- In Figure S3 of the Supporting Information, why does only one phosphorus signal appear despite the presence of two chemically inequivalent phosphorus environments in complex 2? Moreover, please provide the non-decoupling 31P NMR spectrum of 2.
Author Response
The manuscript reports the synthesis and characterization of Cu₁₄H₁₂(PᵗBu₃)₆Cl₂, a novel copper hydride cluster that represents a structural extension of Stryker’s reagent [HCu(PPh₃)]₆, a well-established selective hydrogenation agent. Using SC-XRD, ESI-MS, NMR, and a convolutional neural network combined with DFT calculations, the authors elucidate its architecture: a distorted Cu₈ cube capped by six Cu(PᵗBu₃) units and twelve μ₃-hydrides. The compound demonstrates selective hydrogenation of electron-deficient C=C bonds (e.g., cyclohex-2-en-1-one, dimethyl fumarate) while remaining inactive toward ethylene, closely resembling the reactivity profile of Stryker’s reagent. Overall, this work provides a significant contribution to the development of new structural motifs in multinuclear copper hydride chemistry and highlights its potential as an expanded analogue of Stryker’s reagent. The manuscript is suitable for publication in Molecules after addressing the following minor points:
Answer: Thanks you for the overall positive assessment of our work.
- Page 6: The authors evaluate the reactivity of compound 2; however, the reaction scheme should be included in the main text. Additionally, a direct comparison of its reactivity with reported Stryker’s reagent analogues is necessary.
Answer: Thanks for pointing this out and we added the reaction schemes for both reactions carried out. Additionally, we added a sentence to compare the reactivity between 2 and Stryker’s reagent which we did by comparing the reaction conditions of the reaction with cyclohex-2-en-1-one and the hydrogenation reagent.
- Page 8: The reported yield of compound 2 is quite low. What is the major product of the reaction? Please provide data on the crude mixture in the Supporting Information.
Answer: This is an interesting question that we are currently investigating. We have isolated two additional copper hydrides, which are beyond the scope of this work and are currently being characterized. The reason for the low yield is the required correct ratio between CuH and CuCl, which is achieved by the sub-stoichiometric addition of L-Selectride. However, due to local variations in stoichiometry, a higher yield was not obtained despite several attempts.
- What is the outcome when (tBu₃P)CuCl reacts with an excess of L-Selectride?
Answer: By using an excess of L-Selectride two products are obtained, which are the copper hydride clusters mentioned above. However, in contrast to the relatively stable Cu₁₄, both compounds are more reactive and form elemental copper and hydrogen at higher temperatures, which is actually under investigation.
- In Figure S3 of the Supporting Information, why does only one phosphorus signal appear despite the presence of two chemically inequivalent phosphorus environments in complex 2? Moreover, please provide the non-decoupling 31P NMR spectrum of 2.
Answer: Thanks for pointing this out. Within 2 all phosphines are chemically equivalent. The two signals in the ¹H NMR spectrum are due to coupling of the hydrogens with phosphorus. This has been confirmed by NMR measurements at different field strengths. We have provided the non-decoupled ³¹P NMR spectrum to the Supporting Information.